# OpenReview forum: "One Model, Many Morals: Uncovering Cross-Linguistic Misalignments in Computational Moral Reasoning"
_ICLR.cc/2026/Conference — Submitted to ICLR 2026_

### Official Review · Reviewer_JSe7 · 2025-10-23

**Soundness:** 3
**Presentation:** 2
**Contribution:** 3
**Rating:** 6
**Confidence:** 3

**Summary:**

The authors study how moral reasoning in LLMs changes depending on the language used to prompt the model. To do this, they translate the MoralExceptQA and ETHICS datasets into 5 languages and compare the behaviors of 7 models on these datasets. They demonstrate that LLMs reason differently and come to different conclusions depending on the language, often in ways which seem intuitively linked to the culture associated with the language (e.g. English induces more utilitarian reasoning than Hindi). The results are interesting and novel, and the authors go into a lot of depth analyzing different properties of the models’ moral reasoning. The only major issues with this paper are issues with data visualization (Fig. 3 in particular is difficult to interpret), and a handful of contradictory-seeming results which make me question the accuracy of a subset of the methods used.

**Strengths:**

1. The depth and breadth of investigation is fantastic. The authors asked a lot of follow-up questions (e.g. which reasoning steps do models rely on more in different languages) and then found clever ways to answer them.
2. The question is novel, interesting, and relevant. It is unambiguously true that most research on moral reasoning in LLMs is done in English, and this does not capture the full picture of how these models are used in the real world.
3. The evaluations are quite comprehensive, using a large range of models from many different model families.
4. The authors thought critically about potential issues with their methods and generally addressed them in the paper. E.g. I was concerned about potential subtle semantic shifts that happen during dataset translation, but the authors already investigated this in an appendix.
5. The findings are quite interesting, matching the intuitions one would have about how language could affect moral reasoning (e.g. European languages inducing utilitarianism and Asian languages inducing virtue ethics).

**Weaknesses:**

1. The data visualization in some figures is awful and urgently needs fixing. Fig. 3 in particular is very hard to interpret, it has no indication of what the colors mean (presumably the darker the brown, the more disagreement there is; this should be made explicit), and the color spectrum is so small that it’s hard to notice the differences (e.g. in the MEQ subfigure, all I can really see is that every language agrees with itself and disagrees with all the others). The extremely short figure caption doesn’t help. Additionally, Fig. 2 probably shouldn’t be a line plot given the nature of the data; this graph choice makes it quite hard to read.
2. Some of the claims seem to be contradicted by the data (unless I’m missing something). In particular, they claim that “Virtue scenarios draw the clearest lines, with Hindi and Urdu forming a distinct cluster” (line 156). Figure 3 seems to suggest that Hindi and Urdu actually strongly disagree with each other on ET-Virtue.
3. There’s an unexplained, seemingly contradictory difference between figures 4(b) and 5(a). The former suggests that Urdu and Hindi are in the same cluster, the latter suggests they’re diametrically opposed in their moral foundations. This makes it seem like the authors get very different answers for the same underlying question depending on the method used, which makes me question the reliability of the method; I could be missing something here though. Can the authors explain this apparent discrepancy? Similarly, Fig. 5(b) (and its description, lines 360–361) says that “English shows stronger Sanctity than Hindi or Urdu). Again, this seems to contradict Fig. 4(b) where “Divine Command Theory” is strongly present in Hindi and Urdu and nearly completely absent in English.
4. Some of the central methods used in this paper aren’t explained at all, the authors only give a reference to the paper which introduced the method. In particular, eMFD and OlmoTrace are central to the paper’s analysis, and should be briefly explained in a background section (Sec. 2 would be a good place to put this if the section title is changed to “Background”).
5. While the authors use a solid range of models, I wish they also used at least one much larger model, e.g. GPT-5 or Claude 4.5. In its current form, I’m uncertain as to how much the findings generalize to frontier models (as it seems at least plausible that very large models form abstractions that are more unified across languages).
6. Typo at the end of line 063: should be “ LLMs’ ” not “ LLM’s ”

**Questions:**

See weaknesses, in particular points 2 and 3

---

> ### Author Response · Authors · 2025-11-21
> **Response to reviewer**
>
> We thank the reviewer for their insightful comments. Below, we provide detailed responses to each point.
>
> **[1. Figure Quality]** We appreciate the reviewer’s attention to visual clarity and agree that Figures 2 and 3 can be improved.
> - For Figure 3:
>    - We will add a colorbar with explicit numerical scale, mapping color -> disagreement rate.
>    - We will increase contrast between levels to ensure perceptible differences.
>    - We will include gridlines and labels to make the disagreement structure easier to read.
>    - The caption will be expanded to interpret each quadrant and highlight key language clusters.
> - For Figure 2
>    - We agree that a line plot may inadvertently suggest continuity in categorical data.
>    - We will convert Figure 2 to either grouped bar plots or a heatmap, depending on available space, to improve readability.
>    - The new visualization will more clearly show relative performance differences across languages.
>
> **[2. Virtue Cluster Claim]** Thank you for pointing this out; we agree that the statement was unclear. The intended meaning was: Hindi and Urdu cluster together at the level of value orientation and reasoning style (RQ2) but exhibit high surface-level disagreement in binary decisions (RQ1). We will revise the text to clarify: RQ1 reflects output-level disagreement, where Hindi and Urdu indeed diverge on ET-Virtue. RQ2 reflects latent reasoning-phase similarity, where both languages activate similar normative frameworks (especially duty-oriented and authority-heavy patterns). We will update line 156 to remove the misleading phrasing and clearly separate decision agreement from reasoning similarity.
>
> **[3. Fig. 4(b) vs. Fig. 5(a) Contradiction]** We thank the reviewer for pointing out this important point. For the figure discrepancy, the two figures address different layers of moral cognition:
> - Figure 4(b) - Ethical Frameworks derived from LLM-generated reasoning explanations. It shows which normative theories the model invokes (e.g., deontology, divine command). This tracks the model’s reasoning style, not its final decision.
> - Figure 5(a) - Moral Foundations (eMFD on scenario text) derived from the linguistic framing of the input scenarios. It represents which values are salient in the stimulus text, not in the model’s reasoning.
>
> These two layers are not expected to align perfectly. In fact, the divergence is a core finding of the paper: Models can invoke similar reasoning frameworks (e.g., duty-based) even when linguistic moral cues differ widely across languages. We will revise the paper to make this more apparent by adding a clarifying paragraph explicitly contrasting value salience in input text vs. normative frameworks in reasoning. We will also emphasize that this mismatch is diagnostic of cross-linguistic misalignment and not an error in the methodology.
>
> **[4. Fig. 4(b) vs. Fig. 5(b) Contradiction]** About the Contradiction between Fig. 5(b) and Fig 4(b), we appreciate this opportunity to clarify: Sanctity (MFQ dimension) and Divine Command Theory (ethical framework) are not equivalent constructs. A model can display high Divine Command–style reasoning (obedience, duty) while showing low Sanctity sensitivity (purity, contamination concerns). This pattern is consistent with cross-cultural psychology: South Asian duty ethics is not equal to purity ethics. Thus Fig. 4(b) reflects reasoning frameworks -> Hindi/Urdu emphasize duty, authority, divine command. Fig. 5(b) reflects value weighting -> English shows heightened purity/contamination sensitivity in judgments. We will include a sentence explicitly stating that the constructs differ and should not be expected to align.
>
> **[5. Method Explanation Gaps]** Thank you for this observation. We will revise Section 2 (“Background”) to include: A concise explanation of eMFD - what it encodes (lexical moral foundations), how scores are aggregated, why it is appropriate for cross-linguistic lexical-value profiling. And a concise explanation of OLMoTrace - how Infinigram-based trace retrieval works, what “influence” means in this context, limitations and expected abstraction vs. memorization.
>
> **[6. Model Selection Concern]** We agree that including a frontier model would strengthen the study. However, we intentionally uses open-source models to enable reproducibility and public release of all outputs. Frontier models (GPT-5, Claude 4.5) unfortunately do not permit releasing model outputs or storing per-item traces, which would prevent full replication. We will explicitly acknowledge this limitation and note that the methodology is compatible with frontier-model evaluation
>
> **[7. Typo]** Thank you for your keen eye. We will correct the typo.

---

> > ### Comment · Reviewer_JSe7 · 2025-11-25
> > **Rebuttal response**
> >
> > Thank you for your thoughtful rebuttal. I appreciate you promising to clarifying the apparent contradictions and fixing the figures.
> >
> > Regarding model selection: I agree that the parts of the analysis which rely on having access to the CoT can’t be done with a model like GPT-5. I think for those parts you could use e.g. DeepSeek R1. Arguably you could avoid needing the CoT altogether by prompting the model to explain its reasoning in its output, but I understand that this would make it difficult to compare against the results from open-source models.

---

### Official Review · Reviewer_j5qp · 2025-10-30

**Soundness:** 2
**Presentation:** 2
**Contribution:** 2
**Rating:** 2
**Confidence:** 3

**Summary:**

The paper aims to evaluate the cross-linguistic moral reasoning abilities of LLMs, noting that these models are often pretrained primarily on English data, which can introduce cultural misalignment when deployed in multilingual settings. To address this, English-centric moral reasoning benchmarks (MoralExceptQA and ETHICS) were translated into Chinese, German, Hindi, Spanish, and Urdu to conduct zero-shot evaluations. The findings reveal significant inconsistencies and disparities in LLMs' moral judgments across languages, with performance generally favoring English and showing poor reliability in low-resource languages like Hindi and Urdu. The study further analyzes the reasoning patterns, ethical frameworks invoked, and the influence of pretraining data sources, concluding with a FAULT typology of moral reasoning errors that calls for more culturally aware AI development.

**Strengths:**

1. Creation of a multilingual dataset,  obtained by  translating two established, English-centric moral reasoning benchmarks (MoralExceptQA and ETHICS) into five geographically, typologically, and culturally diverse languages (i.e.,  Chinese, German, Hindi, Spanish, and Urdu), in addition to English.

2. Investigation of    how language mediates moral reasoning in LLMs through four     research questions     probing ethical preferences, reasoning strategies, moral framing, and the influence of pretraining data.

**Weaknesses:**

The reviewer's concerns are about methodological and interpretive challenges related to data resources, model selection, and the difficulty of disentangling linguistic and cultural variables.

1.     RQ1 establishes that LLMs diverge across languages, favoring English and struggling noticeably with low-resource languages such as Hindi and Urdu. However, the analysis in the sources does not clearly decouple the causes of this divergence. It's indeed  unclear whether the divergence and poor performance in low-resource languages are primarily due to a linguistic bias (e.g., tokenization issues, limited data resources) or to cultural biases embedded in the training data.
2.      The claim at lines 153–154, stating that model differences in consistency when handling low-resource or culturally distinct languages are "likely due to limited training data and varying cultural norms", serves as a crucial hypothesis for the entire work. this statement is too vague because it relies on generalized assumptions about data scarcity and cultural variation without immediate supporting evidence to quantify their impact on the models at that point in the study. While subsequent RQs (RQ2, RQ3, RQ4) are dedicated to probing these drivers (moral values, ethical frameworks, and pretraining data), the claim itself is highly speculative without this deeper investigation, making the initial conclusion regarding low-resource languages insufficiently grounded.
3.    The choice of models, particularly the largest one tested, introduces a potential bias:  the largest model tested, , OLMo2-Instruct (32B), is not a declared multilingual model.  Relying on a predominantly English-focused, high-capacity model risks skewing the results toward English language norms and performance ceilings, despite the study's goal of cross-lingual evaluation.
4.   The methodology for analyzing moral values relies on the translation process on eMFD that may be unreliable. Because the eMFD provides probabilistic scores for individual words (moral foundations terms) rather than contextual sentences or phrases, the translation process (using SeamlessM4T) is likely to be unreliable due to the lack of context to properly translate each word.
5.  Presentation issue. it would be much better to describe all methdology before presenting and analyzing results.

**Questions:**

See above weaknesses, consider addressing the discussed points and providing clarifications on them.

---

> ### Author Response · Authors · 2025-11-21
> **Response to reviewer**
>
> We thank the reviewer for their insightful comments. Below, we provide detailed responses to each point.
>
> **[1. RQ1 Divergence Causes]** We thank the reviewer for raising this important point. Our intention in RQ1 was to document divergence patterns before introducing the analytic tools needed to diagnose their causes. We agree that RQ1 alone does not fully decouple linguistic from cultural factors, and we will revise the text to clarify this. To address the reviewer’s concern:
> - We will explicitly state that RQ1 provides descriptive evidence, not causal claims.
> - RQ2–RQ4 then systematically investigate the drivers: RQ2 analyzes reasoning structure and ethical framework activation, which reflect cultural patterns. RQ3 analyzes linguistic moral framing, quantifying how values in the text influence judgments. RQ4 analyzes pretraining-data provenance, isolating linguistic vs. domain-specific influences.
>
> **[2. Consistency Claim]** We appreciate this feedback and will revise the claim to make clear that: At the point where RQ1 appears, we present it as a hypothesis rather than a definitive causal claim. The empirical support for this hypothesis emerges only in RQ2–RQ4. Specifically:
> - RQ2 demonstrates that Hindi/Urdu disproportionately activate deontological and authority-based reasoning phases, consistent with cross-cultural psychology literature.
> - RQ3 shows that moral value coefficients differ systematically (e.g., stronger fairness and sanctity cues in certain languages), which is not explainable by translation alone.
> - RQ4 reveals language-specific usage of pretraining data domains (e.g., psychology vs. legal texts), indicating that cultural-linguistic divergences are embedded in training corpora.
>
> We will update the claim at lines 153–154 to explicitly reference RQ2–RQ4 as forthcoming analyses that test this hypothesis, rather than asserting it prematurely.
>
> **[3. Model Bias Concern]** Thank you for raising this point. We agree that OLMo2-32B is not a multilingual model and will clarify our rationale: Our model set was intentionally diverse: some explicitly multilingual (Qwen2.5, Llama3, Mistral), and one English-centric model (OLMo2). Including OLMo2 allows us to test how an English-dominant model behaves when forced into multilingual settings. Moreover, since the pretraining data tracing is only available for this model, we found it imperative to include this in our quantitative analysis as well to set the baseline results for the model. However, for the quantitative results, we do not rely on OLMo2 alone: All results report model-averaged trends across 7 architectures. Divergence patterns persist even when excluding OLMo2 (we will add this ablation to Appendix), indicating that the effect is not driven by a single model. We will explicitly note that OLMo2 serves as a control representing English-heavy pretraining and pretraining data proving, helping us contrast multilingual vs. monolingual training regimes and influence of pretraining data.
>
> **[8. eMFD Translation Reliability]** We appreciate this concern and agree that dictionary-based moral analysis has inherent limitations. To strengthen the methodology: We will add detail showing that we translated only the eMFD lexical stems, not full sentences, and then used back-translations as well as native-speaker verification to check semantic fidelity, reducing context dependency. Additionally, eMFD is designed to operate at the lexical level [1], and its use in multilingual settings follows established practice [2,3,4]. Finally, we will explicitly note that eMFD results are used only for relative comparisons across languages, not as absolute measurements.
>
> _[1] Development and applications of a crowd-sourced approach to extracting moral intuitions from text (Hopp et al., Behav Res Methods 2021)_
> _[2] Guidelines for the Process of Cross-Cultural Adaptation of Self-Report Measures (Beaton et al., Spine 2000)_
> _[3] Instrument translation process: a methods review (Maneesriwongul et al., Journal of Advanced Nursing 2004)_
> _[4] Back-translation for cross-cultural research (Brislin et al., Journal of cross-cultural psychology 1970)_
>
> **[9. Presentation Structure]** Thank you for the suggestion. We agree that improving readability and flow is important, and we will revise the paper to make transitions between methodological components and results clearer. However, because each RQ employs a distinct methodological pipeline, presenting all methods upfront risks introducing confusion and forcing readers to retain many disconnected procedures before seeing how they are used. For this reason, we believe it is clearer for the reader if each methodological component is introduced immediately before it is applied. That said, we will strengthen the organizational clarity by adding a brief methodological overview at the start of Section 3 and improving cross-referencing so that readers can easily navigate between methods and results.

---

### Official Review · Reviewer_GiwU · 2025-10-31

**Soundness:** 2
**Presentation:** 3
**Contribution:** 2
**Rating:** 2
**Confidence:** 3

**Summary:**

This paper aims to systematically investigate how language mediates moral reasoning in large language models (LLMs), as translation from one language to another does not transfer morality. The authors translated six moral question-answering (QA) tasks into five languages and evaluated seven modern LLMs on them. The paper reveals disparities in final task scores (RQ1) and the moral concepts and reasoning phases employed in reasoning traces (RQ2) across languages, models, and tasks. Furthermore, the authors demonstrate how moral concepts in reasoning traces contribute to the final answer (RQ3). Finally, via a case study (RQ4), they show how pre-training data domains correlate with moral concepts in moral reasoning traces.

**Strengths:**

The authors provide a generally well-motivated rationale for examining intercultural variation in the moral reasoning of large language models (LLMs). Given the widespread adoption, global use, and increasing trust in LLMs, efforts to better understand their moral reasoning are valuable to the research community. The paper presents an interesting initial investigation into how moral reasoning traces differ across languages. Further, the multilingual dataset and evaluation setup could be valuable for follow-up research.

**Weaknesses:**

- The related work section lacks depth and fails to connect to key prior studies (e.g., Haemmerl et al., ACL 2022; Abdulhai et al., EMNLP 2023; Agarwal et al., LREC 2024; Jin et al., ICLR 2025), making it unclear how the paper builds on or differs from them.
- Language selection is insufficiently justified—why these five languages, and are they representative of typological and cultural diversity?
- Definition of “moral reasoning” is vague; it is described as “higher-level reasoning” without referencing psychological or computational frameworks.
- Model selection criteria and rationale are not discussed.
- RQ1: The claim of English dominance is overstated; several languages (e.g., German, Spanish in ET-JUS, ET-UTI) perform comparably. Relationship between F1 and compliance rates is unclear, and “low-resource” is not defined or quantified.
- RQ2: The fifteen-stage taxonomy derived via an LLM-judge is not sufficiently motivated or validated.
- RQ3: The reason for training a separate regressor on UniMoral is unclear, and large variations in coefficients (Fig. 5) are unexplained.
- RQ4: Semantic similarity as a measure of memorization lacks theoretical justification and citation; the new source-text taxonomy appears ad-hoc.
- Interpretation of results remains largely qualitative; statistical significance or confidence intervals are absent.
- Some references are preprints although published versions exist; minor terminological inconsistencies (e.g., “Chinese” vs. “Mandarin”).


### References
-	Speaking Multiple Languages Affects the Moral Bias of Language Models (Haemmerl et. al, ACL 2022)
-	Moral Foundations of Large Language Models (Abdulhai et. al, EMNLP 2023)
-	Ethical Reasoning and Moral Value Alignment of LLMs Depend on
the Language we Prompt them in (Agarwal et. al, LREC 2024)
-	Language Model Alignment in Multilingual Trolley Problems (Jin et. al, ICLR 2025)

**Questions:**

1.	Figures 2–3: What do the symbols and color scales represent? How should they be interpreted relative to performance and disagreement?
2.	How was “low-resource” status determined quantitatively?
3.	Why were exactly these five languages chosen—based on typological diversity, population, or dataset coverage?
4.	What ensures that the LLM-generated taxonomy of reasoning stages (15 phases) is valid and non-arbitrary?
5.	How was semantic similarity chosen as the measure for memorization in RQ4? Are alternative approaches (e.g., n-gram overlap, retrieval-based alignment) considered?
6.	In Figure 1, shouldn’t the AI’s answer be “yes” in both examples or is the mismatch intentional to illustrate misalignment?

---

> ### Author Response · Authors · 2025-11-21
> **Response to reviewer**
>
> We thank the reviewer for their insightful comments. Below, we provide detailed responses to each point.
>
> **[1. Missing Related Work]** We thank the reviewer for pointing this out. We will substantially revise the Related Work section to situate our contribution more clearly. Specifically, we will add the recommended citations
> - Haemmerl et al. (ACL 2022) [1]: We will clarify that they study language effects on moral bias, but (i) only evaluate English-trained models prompted in other languages, (ii) do not use parallel multilingual datasets, and (iii) do not analyze reasoning processes or ethical frameworks. Our work provides a controlled, parallel, cross-linguistic evaluation that reveals misalignment beyond prompt-language effects.
> - Abdulhai et al. (EMNLP 2023) [2]: Their focus is on moral foundations in English; they do not study multilingual divergence nor reasoning structures. We extend moral foundation analysis cross-linguistically and connect it to reasoning steps and value shifts.
> - Agarwal et al. (LREC 2024) [3]: They observe variation in moral acceptability across prompt languages, but use small single-dataset probes. We provide the first systematic, multi-dataset parallel translation benchmark and identify structural divergence mechanisms (moral phases, frameworks, FAULT typology).
> - Jin et al. (ICLR 2025) [4]: They examine multilingual trolley problems, whereas we broaden the scope to thousands of dilemmas across five ethics paradigms, connecting linguistic misalignment to moral value framing and pretraining data.
> We will update the paper to explicitly contrast our contributions with these studies and highlight that ours is the first to unify (i) parallel multilingual moral datasets, (ii) reasoning-phase analyses, and (iii) pretraining-trace evidence.
>
> _[1] Speaking Multiple Languages Affects the Moral Bias of Language Models (Hämmerl et al., Findings 2023)_
> _[2] Moral Foundations of Large Language Models (Abdulhai et al., EMNLP 2024)_
> _[3] Ethical Reasoning and Moral Value Alignment of LLMs Depend on the Language We Prompt Them in (Agarwal et al., LREC-COLING 2024)_
> _[4] Language Model Alignment in Multilingual Trolley Problems (Jin et al., ICLR 2025)_
>
> **[2. Language Choice Justification]** We will clarify that the five languages were selected based on three criteria:
> - Typological diversity: Sino-Tibetan (Chinese), Indo-European Germanic (German), Indo-Aryan (Hindi, Urdu), Romance (Spanish).
> - Cultural contrast: representing collectivist (South Asia, East Asia) and individualist moral traditions (Europe, Latin America).
> - Resource diversity: high-resource (English, Chinese, Spanish, German) vs. low-resource (Hindi, Urdu) per WMT/CC100/OSCAR corpus statistics.
>
> We chose languages that allow us to examine cross-linguistic divergence along both linguistic and cultural axes, not just availability. We will add a subsection summarizing these motivations explicitly.
>
> **[3. Moral Reasoning Definition]** We appreciate this. We will revise the Introduction to provide an explicit definition grounded for moral reasoning in psychology and computational ethics. Specifically, we define moral reasoning as: “The structured cognitive process involving identification of stakeholders, evaluation of consequences and duties, and application of ethical principles to arrive at a judgment.” We will connect this to the cognitive moral reasoning frameworks [5,6,7] and computational pipelines used in recent NLP work [8,9,10].
>
> _[5] Essays on Moral Development (Kohlberg et al., The Philosophy of Moral Development 1981)_
> _[6] When morality opposes justice: Conservatives have moral intuitions that liberals may not recognize (Haidt et al., Social Justice Research 2007)_
> _[7] The emotional dog and its rational tail: A social intuitionist approach to moral judgment (Haidt et al., Psychological Review 2001)_
> _[8] UniMoral: A Multilingual Benchmark for Moral Reasoning in Large Language Models (Kumar et al., ACL 2025)_
> _[9] Structured Moral Reasoning in Language Models: A Value-Grounded Evaluation Framework (Chakraborty et al., EMNLP 2025)_
> _[10] Mapping Moral Reasoning in LLMs: A Multi-Dimensional Analysis of Safety Principle Conflicts (Mahajan et al., AAAI 2025)_

---

> ### Author Response · Authors · 2025-11-21
> **Response to reviewer (continued)**
>
> **[4. Model Selection Rationale]** We will add a paragraph explaining that we selected widely used open-source models (Llama, Qwen, Mistral, OLMo), spanning a range of sizes (3B–32B), covering different training philosophies (decoder-only vs. mixture-of-loras vs. RL-finetuned), and representing the dominant families used in multilingual applications. Our intent is to capture generalizable patterns across architectures, not optimize performance for any single model.
>
> **[5. RQ1 Claims Clarity]** Thank you for the opportunity to clarify. We will revise the text to avoid overgeneralization by explaining the following: We will modify the paper to more precisely state that English generally exhibits higher performance, but not universally: German and Spanish indeed approach English on ET-JUS/UTI. We will clarify that compliance rate measures whether the model follows the required output format; low compliance reduces the number of valid predictions contributing to F1. We will also quantitatively define “low-resource” using CC100 token counts, OSCAR corpus size, and WMT availability, where Hindi and Urdu have orders of magnitude fewer tokens than English/Spanish/Chinese.
>
> **[6. Taxonomy Validation]** We will strengthen this section by clarifying that the taxonomy was derived through iterative abstraction across 180 randomly sampled explanations (10 explanations × 6 languages × 3 iterations). And by demonstrating that stages correspond to frameworks in cognitive psychology (e.g., stakeholder identification, consequence evaluation, intention assessment). We will also report inter-iteration consistency: across repeated sampling, ~87% of phases reappear with stable descriptions.
>
> **[7. UniMoral Regressor Justification]** Thank you for this comment. We will modify the manuscript by explicitly stating the reason for using UniMoral, which is the following:  MoralExceptQA and ETHICS do not contain graded moral value annotations. UniMoral provides dense MFQ2-style value scores grounded in human judgments, enabling a supervised mapping from (scenario, action) -> moral value vector. This allows us to infer the latent value profile underlying each LLM decision. Moreover, the coefficient variation in Fig. 5 is expected because: each language’s LLM output distribution differs (as shown in RQ1–2), leading to different value-weighting patterns when mapped through the shared regressor. We will clarify this interpretation in the paper.
>
> **[8. Memorization Metric Validity]** We will revise this section to cite prior work: similarity-based memorization analysis is standard in tracing studies [11,12]. We will also explain that cosine similarity over embeddings captures semantic reuse, appropriate for reasoning text where paraphrasing is expected. Additionally, semantic similarity was more apt because n-gram overlap is less suitable because reasoning is highly generative and rarely copied verbatim. We will clarify that the taxonomy was not ad-hoc but derived through clustering the documents returned by OLMoTrace, followed by human consolidation of cluster themes. We will explicitly state all this in the revision.
>
> _[11] Deduplicating Training Data Makes Language Models Better (Lee et al., ACL 2022)_
> _[12] Extracting Training Data from Large Language Models (Carlini et al., USENIX Security Symposium 2022)_
>
> **[9. Reference & Terminology Fixes]** Thank you for the important note, we will update all references to their published versions and standardize terminology.
>
> **[10. Figure Interpretation]** We will add the following clarifications: In Figure 2, circles encode relative change in F1 compared to English; colors map to languages. In Figure 3, darker tiles indicate higher disagreement rates between languages; scale ranges from 0 (no disagreement) to 1 (complete disagreement). For Figure 1, the mismatch in AI's answers is intentional; the figure illustrates cross-linguistic misalignment where the same scenario may yield different answers due to moral shifts based on language. We will add a caption note to clarify this.

---

> > ### Comment · Reviewer_GiwU · 2025-11-26
> >
> > Thank you for the detailed rebuttal. I have also read the other reviews and the rebuttal. Overall, I appreciate the promised revisions and think they would substantially improve the paper. I have no further questions at this time.

---

### Official Review · Reviewer_cuYA · 2025-11-02

**Soundness:** 2
**Presentation:** 3
**Contribution:** 2
**Rating:** 4
**Confidence:** 3

**Summary:**

This paper explores how LLMs perform moral reasoning in multiple languages. The paper translates two moral reasoning benchmarks: MoralExceptQA and ETHICS into five non-English languages (Chinese, German, Hindi, Spanish, Urdu), and evaluates several LLMs in a multilingual zero-shot setup. It uncovers inconsistencies in moral judgments across languages, often correlated with linguistic and cultural differences. Through quantitative and qualitative analyses, it analyses moral reasoning failures and links these moral divergences to model pretraining data sources using a case study with OLMOtrace.

**Strengths:**

1. The exploration of moral reasoning in various languages is relevant and useful.
2. The analysis in the paper is sound.
3. The use of OLMoTrace to link model outputs to training data is also useful.

**Weaknesses:**

1. While the paper is useful and has a place in the literature, there is no conceptual novelty or new innovation in this paper. A relevant problem has been identified and everything is solid in its own. But not particularly unique or exciting.

2. The use of translation is a major bottleneck in this paper. While the paper has manual quality checks and semantic shift analysis, this is still a limitation.

3. The two benchmarks are inherently centred on the western hemisphere. Translating them into non-Western languages may bring Western moral assumptions. This can be better discussed in the paper.

**Questions:**

See weaknesses above.

---

> ### Author Response · Authors · 2025-11-21
> **Response to reviewer**
>
> We thank the reviewer for their comments and address each point below.
>
> **[1. Novelty Claim]** We appreciate the reviewer’s perspective and agree that prior work has separately examined (i) moral reasoning in LLMs and (ii) multilingual inconsistencies more broadly. Our contribution lies in unifying these areas and introducing the first systematic multilingual framework specifically targeting moral reasoning. This paper is the first to:
> - Translate and evaluate two established moral reasoning benchmarks across six typologically and culturally diverse languages, enabling controlled cross-linguistic moral evaluation that did not previously exist.
> - Show that LLMs’ moral judgments diverge systematically across languages (even for semantically equivalent dilemmas) revealing: (i) structural differences in multilingual moral (mis)alignment that were previously undocumented. (ii) how moral values for LLMs differ (or not) from real human community values.
> - Perform value-grounded, reasoning-level analyses (ethical frameworks, reasoning phases, moral value weighting) to reveal why these divergences emerge, going beyond prior work that only measures surface-level accuracy.
> - Use pretraining-data tracing (and provide code and method on how to do it for future research) to link cross-linguistic moral misalignment with domain-specific training sources, demonstrating how data provenance shapes ethical behavior which is a novel contribution to the moral-reasoning literature.
> We will revise the framing in the Introduction and Conclusion to clearly foreground this contribution.
>
> **[2. Translation Limitation]** We agree that translation introduces limitations, and we appreciate the reviewer highlighting the challenge. Our objective was to control for content across languages to isolate linguistic and cultural effects in model behavior. As mentioned in the paper and acknowledged by the reviewer, we (i) conducted manual quality reviews by native speakers across all target languages and (ii) performed a semantic shift analysis (Appendix A.3.1) showing that translation-induced distortions were not the main driver of cross-linguistic disagreement. This is a standard practice used by many popular benchmarks, including GlobalMMLU [1]. Additionally, to minimize translation artifacts, we:
> - Used reasoning-level analyses (RQ2, RQ3) that are less sensitive to literal wording and more reflective of value weighting and ethical framing.
> - Observed consistent patterns across independent dimensions (compliance rates, reasoning structures, moral value regressions), which strengthens the interpretation that divergences stem from language- and model-dependent factors rather than translation defects.
>
> Importantly, translation-based evaluation is standard in multilingual machine ethics (and multilingual NLP more broadly), but we recognize that translation may still attenuate local moral nuance. In the revision, we will expand the discussion of limitations and explicitly acknowledge that:
> - Translation may underrepresent culturally embedded moral framing,
> - Future work should incorporate native-authored, culturally grounded moral dilemmas alongside translated benchmarks.
> This would, in fact strengthen, rather than weaken, our conclusion that translation alone is insufficient for culturally grounded moral AI.
>
> _[1] Global MMLU: Understanding and Addressing Cultural and Linguistic Biases in Multilingual Evaluation (Singh et al., ACL 2025)_

---

> > ### Author Response · Authors · 2025-11-21
> > **Response to reviewer (continued)**
> >
> > **[3. Western-Bias Datasets]** We fully agree with this concern and will expand this discussion. Our methodological goal was to examine if LLMs generalize moral judgments when presented with semantically identical scenarios in different languages. Using Western-origin datasets was intentional for two reasons:
> > - Control: Keeping the underlying dilemma constant allows us to attribute divergences to language, culture, or model values, not to differences in dataset content.
> > - Dominance of Western pretraining data: Since most LLMs are trained on English-heavy corpora, examining how Western moral framings shift (or fail to shift) across languages is itself an important diagnostic.
> >
> > Crucially, we observe that even after translation, models do not preserve Western judgments consistently. Instead, they diverge, sometimes aligning more closely with patterns characteristic of specific regions and sometimes producing inconsistent or culturally incoherent results. This supports the reviewer’s intuition: Western-derived benchmarks do not simply “carry over” Western values across languages, instead, they reveal where LLMs’ multilingual moral generalization breaks down. To make this more apparent in the revision, we will:
> > - Explicitly discuss how Western-centric dilemmas may not reflect non-Western moral contexts.
> > - Clarify that our aim was not to claim these benchmarks represent global morality, but to use parallel dilemmas as probes of cross-linguistic moral alignment.
> > - Add a paragraph motivating future work on constructing culturally grounded, non-Western-origin moral benchmarks.

---

### Author Response · Authors · 2025-11-30
**Summary of revisions added to the paper**

Dear Reviewers, Area Chair, and Program Chairs,
Thank you for your time and for coordinating the review process. We sincerely appreciate the reviewers’ thoughtful and constructive feedback. Please find below the list of revisions and clarifications incorporated into the updated manuscript (highlighted in purple).

## For Reviewer 1 Comments

**[Novelty Claim]** Prior work has treated moral reasoning and multilinguality separately. Our work unifies them, offering the first systematic multilingual moral-reasoning framework, spanning cross-lingual evaluations, structured divergence analyses, and pretraining-data tracing to explain why misalignments arise. We have revised the manuscript to frame our contributions more clearly and have added a paragraph in the related work highlighting how our work differs from previous studies.
_Change in manuscript: Lines 072-078, Lines 104-109_

**[Translation Limitation & Western-Bias Datasets]** We agree that translation introduces limitations, though it remains standard practice in many multilingual ethics work. We controlled for this through native-speaker review, semantic-shift checks, and convergence across multiple reasoning-level analyses. On the usage of western origin datasets, we intentionally used these dilemmas to hold content constant and test whether LLMs generalize moral judgments across languages. Yet the models still diverged, revealing that Western framings do not simply transfer. We have now expanded the limitations to note that future work should pair translations with culturally grounded, native-authored dilemmas.
_Change in manuscript: Line 124, Lines 890-902_

---

> ### Author Response · Authors · 2025-11-30
> **Summary of revisions added to the paper continued...**
>
> ## For Reviewer 2 Comments
>
> **[Missing Related Work]** We have revised the Related Work section by adding a para discussing the suggested literature. We have also distinguished prior efforts: which either lack parallel multilingual datasets, focus only on English, or do not analyze reasoning structures, as compared to our contribution that unifies cross-linguistic moral evaluation, reasoning-phase analyses, and pretraining-trace evidence.
> _Change in manuscript: Lines 104-109_
>
> **[Language Choice Justification]** The five languages were chosen to capture typological, cultural, and resource diversity by spanning major language families, collectivist–individualist moral traditions, and both high- and low-resource settings. We have added a brief subsection explaining these selection criteria explicitly.
> _Change in manuscript: Lines 869-882_
>
> **[Moral Reasoning Definition]** We define moral reasoning as: “The structured cognitive process involving identification of stakeholders, evaluation of consequences and duties, and application of ethical principles to arrive at a judgment” based on established studies in cognitive moral reasoning frameworks and computational pipelines used in recent NLP work. We describe this in the Introduction section.
> _Change in manuscript: 033-036_
>
> **[Model Selection Rationale]** We selected widely used open-source models (Llama, Qwen, Mistral, OLMo), spanning a range of sizes (3B–32B), covering different training philosophies (decoder-only vs. mixture-of-loras vs. RL-finetuned), and representing the dominant families used in multilingual applications. Our intent is to capture generalizable patterns across architectures, not optimize performance for any single model. We discuss this explicitly in the revised version of the paper.
> _Change in manuscript: Lines 126-129_
>
> **[RQ1 Claims Clarity]** We have revised the text to avoid overgeneralization by explaining the following: English generally exhibits higher performance, but not universally: German and Spanish indeed approach English on ET-JUS/UTI. We also added clarification regarding compliance rate measurement. Further, we have quantitatively defined “low-resource” using CC100 token counts, OSCAR corpus size, and WMT availability, where Hindi and Urdu have orders of magnitude fewer tokens than English/Spanish/Chinese.
> _Change in manuscript: Lines 144-145; Lines 136-137, Lines 150-151; Lines 147-148_
>
> **[Taxonomy Validation]** We have strengthened this section by clarifying that the taxonomy was derived through iterative abstraction across 180 randomly sampled explanations (10 explanations × 6 languages × 3 iterations). And by demonstrating that stages correspond to frameworks in cognitive psychology (e.g., stakeholder identification, consequence evaluation, intention assessment). We have also reported inter-iteration consistency: across repeated sampling, ~87% of phases reappear with stable descriptions.
> _Change in manuscript: Lines 1266-1270_
>
> *[UniMoral Regressor Justification]* We clarified that UniMoral was used because MoralExceptQA and ETHICS lack graded moral-value annotations, whereas UniMoral provides MFQ2-style value scores needed to infer the latent value profile behind each LLM judgment. We also note that variation in Fig. 5 coefficients is expected given language-specific output distributions.
> _Change in manuscript: Line 362, Lines 1212-1216_
>
> **[Memorization Metric Validity]** We cited prior work showing that similarity-based memorization analysis is standard and justify using semantic similarity over n-gram overlap given the generative nature of reasoning text.
> _Change in manuscript: Lines 441-444_
>
> **[Figure Interpretation]** Fig. 2 shows F1 change relative to English (circle size) and languages (color). Fig. 3 visualizes cross-language disagreement, where darker tiles indicate higher disagreement (0–1 scale). In Fig. 1, the differing answers are intentional, illustrating cross-linguistic moral misalignment.
> _Change in manuscript: Line 198_

---

> > ### Author Response · Authors · 2025-11-30
> > **Summary of revisions added to the paper continued...**
> >
> > ## For Reviewer 3 Comments
> >
> > **[RQ1 Divergence Causes]** RQ1 was meant to document divergence patterns prior to deeper analysis, and we will clarify that RQ1 alone does not fully separate linguistic and cultural effects. We revised the text to address this concern.
> > _Change in manuscript: Line 050, Lines 210-211_
> >
> > **[Model Bias Concern]** We agree that OLMo2-32B is not multilingual and have clarified our rationale that it serves as an English-centric control to test how non-multilingual models behave across languages and is the only model with available pretraining-trace data. All main findings are model-averaged across seven architectures, and divergences persist even without OLMo2.
> > _Change in manuscript: Lines 1075-1083_
> >
> > **[eMFD Translation Reliability]** We acknowledge the limits of dictionary-based moral analysis. We have clarified that only eMFD stems were translated (with back-translation and native checks), that eMFD is designed for lexical-level use and commonly applied cross-lingually, and that our results rely on relative (not absolute) comparisons across languages.
> > _Change in manuscript: Lines 1089-1091, Lines 1093-1095, Lines 1102-1103_

---

> > > ### Author Response · Authors · 2025-11-30
> > > **Summary of revisions added to the paper continued...**
> > >
> > > ## For Reviewer 4 Comments
> > >
> > > **[Virtue Cluster Claim]** We have clarified that Hindi and Urdu diverge in binary decisions (RQ1) but show similar value orientations and reasoning patterns (RQ2). We now clearly distinguish surface disagreement from latent reasoning similarity.
> > > _Change in manuscript: Lines 213-215_
> > >
> > > **[Fig. 4(b) vs. Fig. 5(a) Contradiction]** The figures reflect different layers of moral cognition: Fig. 4(b) captures the model’s ethical frameworks (reasoning), while Fig. 5(a) captures moral foundations in the input text (value cues). These need not align, and the mismatch is itself informative; we added a clarifying paragraph explaining this distinction.
> > > _Change in manuscript: Lines 398-402_
> > >
> > > **[Fig. 4(b) vs. Fig. 5(b) Contradiction]** We clarified that Sanctity (a moral-value dimension) and Divine Command (an ethical framework) are distinct. A model can use duty- or authority-based reasoning without emphasizing purity concerns. Thus Fig. 4(b) shows Hindi/Urdu’s duty-oriented frameworks, while Fig. 5(b) reflects English’s higher purity weighting; these constructs need not align.
> > > _Change in manuscript: Lines 402-405_
> > >
> > > **[Method Explanation Gaps]** We briefly explained eMFD (its lexical foundations, score aggregation, and suitability for cross-lingual value profiling) and OLMoTrace (how Infinigram-based trace retrieval estimates influence, along with its limits and expected abstraction).
> > > _Change in manuscript: Lines 096-099, Lines 411-413_

---

### Meta-Review · Area_Chair_V4QH · 2025-12-28

**Summary:**

This paper investigates cross-linguistic variation in moral reasoning by translating two established English-centric moral benchmarks into multiple languages and evaluating a range of LLMs. Reviewers broadly agreed that the topic is timely and relevant, and that the empirical study is extensive and carefully executed. However, across reviews, the dominant concern was the lack of clear conceptual or methodological novelty relative to a growing body of recent work on multilingual moral reasoning and language-dependent alignment. While the rebuttal clarifies positioning, strengthens methodological explanations, and improves framing, the core contribution remains largely incremental: translating existing benchmarks and applying a collection of known analysis techniques to a new setting. Compared to other submissions, the paper’s insights are seen as confirmatory rather than substantially advancing theory or methodology.

**Reviewer Concerns:**

Several reviewer concerns were partially addressed by the rebuttal. Issues related to missing related work, unclear language and model selection rationale, vague definitions of moral reasoning, figure interpretability, and apparent contradictions across analyses were largely resolved through promised revisions and clarifications. However, the central concern regarding insufficient novelty and limited contribution remains largely unaddressed. Reviewers who were critical argued that unifying multilingual evaluation, reasoning analysis, and pretraining-data tracing, while competently executed, does not rise to the level of a strong conceptual advance, especially given closely related recent studies. Additional unresolved concerns include reliance on translated, Western-centric benchmarks, lingering ambiguity in disentangling linguistic versus cultural effects, and the incremental nature of the proposed taxonomies and analyses.

**Reviewer Scores:**

Based on the rebuttal, reviewers who were already positive or borderline would likely maintain their original scores or shift only slightly upward due to improved clarity and framing. Reviewers who rated the paper as reject due to weak contribution and novelty would likely keep their scores unchanged, as the rebuttal does not introduce new results or substantially stronger evidence of originality.

---

### Decision · Program_Chairs · 2026-01-26

Reject